# Effect of Sulfur and Urea Fortification of Fresh Cassava Root in Fermented Total Mixed Ration on the Improvement Milk Quality of Tropical Lactating Cows

**DOI:** 10.3390/vetsci7030098

**Published:** 2020-07-23

**Authors:** Chanadol Supapong, Anusorn Cherdthong

**Affiliations:** Tropical Feed Resources Research and Development Center (TROFREC), Department of Animal Science, Faculty of Agriculture, Khon Kaen University, Khon Kaen 40002, Thailand; chanadol@kkumail.com

**Keywords:** rumen fermentation, blood thiocyanate, milk thiocyanate, somatic cell count

## Abstract

The aim of the present research was to determine the influence of sulfur and urea combined with fresh cassava root in fermented total mixed ration (FTMR) on digestibility, fermentation in the rumen, blood metabolite, milk yield, and milk quality in tropical lactating dairy cows. Four mid-lactation Thai Holstein–Friesian crossbred cows were studied. Pre-experiment milk yield was 12.7 ± 0.30 kg/day, and the body weight was 495 ± 40.0 kg. Animals were evaluated in a 2 × 2 factorial in a 4 × 4 Latin square design to receive diets followed by: factor A, which was a dose of sulfur inclusion at 1.0% and 2.0%, and factor B, which was level of urea inclusion at 1.25% and 2.5% DM in FTMR. The hydrogen cyanide (HCN) concentrations reduced 99.3% to 99.4% compared with fresh cassava root when FTMR was supplemented with 1.0% and 2.0% sulfur, respectively. Intake of crude protein was increased based on urea level addition (*p* < 0.05). Blood thiocyanate concentration was increased by 21.6% when sulfur was supplemented at 2.0% compared to 1.0% (*p* < 0.05). There was no difference in protozoal concentration, whereas bacterial populations at 4 h after feeding were significantly greater by 6.1% with the FTMR supplemented with 2.0% sulfur and 2.5% urea (*p* < 0.01). Allantoin concentrations, excretion, absorption, and microbial crude protein showed significant interactions between sulfur levels and urea levels in cows fed diets supplemented with 2.0% sulfur and 2.5% urea (*p* < 0.05). The molar ratios of the volatile fatty acid (VFA) profile were affected by dietary FTMR (*p* < 0.01). Furthermore, propionic acid increased by 4.6% when diets were supplemented by 2.5% sulfur (*p* < 0.01). Milk fat and total solids increased when feed was supplemented with 2.0% sulfur and 2.5% urea (*p* < 0.05). The diets supplemented with 2.0% sulfur levels resulted in greater concentrations of milk thiocyanate (*p* < 0.05). The somatic cell count was significantly reduced throughout the experiment with increasing sulfur supplementation (*p* < 0.05). Animals fed diets supplemented with 2.0% sulfur exhibited a decreased somatic cell count by 18.3% compared with those fed diets supplemented with 1.0% sulfur. Thus, inclusion of 2.0% sulfur with 2.5% urea in FTMR containing fresh cassava root improved digestibility, ruminal fermentation, microbial crude protein synthesis, and milk qualities in dairy cows.

## 1. Introduction

Fresh cassava root as an energy source is of interest to supplement in animal diets because it is available in large volumes in tropical zones and is gradually becoming ubiquitously utilized in ruminant diets in tropical areas [1]. The HCN in the cyanogenic fresh cassava root contains values of hydrocyanic acid in the range of 90–114 ppm [2]. HCN has severe cytotoxic effects in the mitochondrial respiratory chain due to its inhibition of cytochrome oxidase [3]. In ruminants, it is dangerous and usually will cause death when an animal receives HCN at a concentration of 1000 ppm (dry matter [DM] basis) [4]. Sulfur can be fed directly in the diet; microbial use of sulfur-containing amino acids synthesis is essential for maximum microbial growth and DM digestibility and can reduce the concentration of HCN [5].

Supplementing non-protein nitrogen to animal diets is to provide ammonia-nitrogen (NH_3_-N) that might be incorporated in their proteins by the rumen microorganisms. Due to the fact that microbial rumen needed supplemental sulfur when feeding non-protein nitrogen to completely utilize NH_3_-N for protein synthesis [1,6]. Sulfur and nitrogen are combined to improve the synthesis of amino acids, and digestion of cellulose by rumen microbes through higher activity of fiber fermentation [7].

Milk thiocyanate is oxidized to yield hypothiocyanite and hypothiocyanous acid and the iodide ion is oxidized to produce hypoiodite and hypoiodous acid, which has efficient antimicrobial properties [8]. Lactoperoxidase has the ability to induce antibacterial effects by acting against the mode of action of cytoplasmic enzymes to damage the cell bacterial membranes and reduce growth [9]. These catalyze the oxidation of thiocyanate by hydrogen peroxide through the synthesis of the antibacterial hypothiocyanite and other substrates [10]. Hydrogen cyanide (HCN) was reported to be changed to innocuous thiocyanate by the action of the rhodanese enzyme. The rhodanese enzyme is synthesized in the liver and kidneys of animals; it participates in detoxification of HCN to thiocyanate [3]. Thiocyanate is transported mainly via urine, milk, and saliva for elimination from the body [11].

Study on fresh cassava root as an energy source in FTMR for topical lactating cows is limited; particularly, nobody has elucidated the result on lactating cow’s work into the effect of the HCN on the somatic cell count. It was hypothesized that FTMR containing fresh cassava root with sulfur and urea addition improves feed efficiency, rumen fermentation efficiency, and milk quality. The aim of the present research was to determine the influence of sulfur and urea combined with fresh cassava root in fermented total mixed ration (FTMR) on digestibility, fermentation in the rumen, blood metabolite, milk yield, and milk quality in tropical lactating dairy cows.

## 2. Materials and Methods

All methodology involving animal care and arrangement were approved by Committee of Animal Experimentation and were performed under the Institutional Guidelines of Khon Kaen University, National Research Council of Thailand (record no. ACUC-KKU 32/61).

### 2.1. Animal and Facility Details

A low number of animals were used in this study due to the limited number of cows with the same condition (DIM, BW, and lactation, etc.) available in the research station. Lowering an error by man, sample collection, and animal managements were carefully performed. This experiment was conducted from October 2018–January 2019, which is the winter season in Thailand. The temperature was about 25–32 °C and did not have much effect on animal stress. Cows were housing in the research station of the institute, which is specific to arrange to do the research. Thus, based on the facility, rumination behavior and digestibility were not influenced by the environmental effect. Cows were in individual pens (5 × 5 m^2^) of the same room in a free-stall environment throughout the study, where they had visual or auditory cues from other cows. Rubber mats were provided as bedding, which replaced every 21 days and cleaned every morning. Before starting the experiment, the animals were trained-in with an individual pen for at least 2 weeks, so the animals could adapt well to individual pens and start an experiment later. Animals were healthy throughout the study.

### 2.2. Cows, Feed and Treatments

Four mid-lactation Thai Holstein–Friesian crossbred cows were studied. Pre-experiment milk yield was 12.7 ± 0.30 kg/day, and the body weight was 495 ± 40.0 kg. Animals were evaluated in a 2 × 2 factorial in a 4 × 4 Latin square design to receive diets followed by: factor A, which was a dose of sulfur inclusion at 1.0% and 2.0%, and factor B, which was level of urea inclusion at 1.25% and 2.5% DM in FTMR. All cows had ad libitum access to FTMR that was provided fresh twice daily (07:00 and 16:00), allowing for 10% refusals. Refusals were collected and weighed each morning and afternoon before fresh feed was provided. All cows were kept in individual pens of the same room, which they had visual or auditory cues from other cows. Clean fresh water was available at all times. The study involved four periods, each period consisting of 21 days. Feed intake was recorded for 21 days, and the last 7 days were dedicated to collection of samples of rumen fluid, feces, and blood. Fresh cassava root variety Kasetsart 50 in the diet was passed through a machine (Yasothon Thailand) to sieve 1 cm. Fermented total mixed ration was made by packing concentrate mixed with rice straw chopped to a length of 3–5 cm by machine and adding water to achieve a moisture content of 55% in processing, as per the process carried out by Supapong et al. [12]. Fermented total mixed ration was then packed in plastic bags to ferment under anaerobic conditions and stored indoors (25–32 °C) for 7 days before use. Table 1 presents the composition of experimental diets.

### 2.3. Sample Collection and Chemical Analysis

The experimental feeds, feces and urine were sampled during the last day of each experimental period. The feces was obtained by rectal sampling and spot sampling was used to collect urine. The feeds and feces were oven dried at 60 °C; each sample was ground and analyzed using methods by the AOAC [13] for DM, crude protein (CP), and ash. Digestibility of nutrients was determined by acid-insoluble ash (AIA) [14]. The method of Van Soest et al. [15] was used to analyze neutral detergent fiber (NDF) and acid detergent fiber (ADF). Nutrient digestibility was analyzed using Van Keulen and Young [14] equation. The pH was immediately recorded with a hand machine (HANNA Instruments HI 8424 microcomputer, Singapore). Fresh cassava root and FTMR were evaluated following Bradbury et al. [16] to analyze HCN concentrations.

The samples of urine were examined for allantoin and creatinine concentrations using high-performance liquid chromatography (HPLC) and for microbial purines absorbed, microbial nitrogen, and microbial crude protein (MCP) according to Chen and Gomes [17] equation. The efficiency of microbial nitrogen synthesis (EMNS) was evaluated using the equation of Agricultural Research Council (ARC) [18]. Metabolizable energy (ME) was determined using Robinson et al. [19] equation.

Milk yields were sampled every day for each animal. Milk samples were collected two times per day (at 05:00 and 16:00) and mixed together for analysis of fat, protein, lactose, total solids, and solids-not-fat content using Milko-Scan33 (Foss Electric, Hillerod, Denmark). The somatic cell counts were analyzed using the Fossomatic 5000 Basic. In addition, milk thiocyanate was determined by Jacob et al. [20] method.

Blood samples were collected from the jugular vein; 8 mL at 0 h and 4 h post morning feeding at day 21 of each period for analysis of blood urea nitrogen (BUN) [21] and blood thiocyanate according to Lambert et al. [22]. Blood was collected in plasma tubes. Ethylenediaminetetraacetic acid (12 mg) was added as an anticoagulant and centrifuged at 500× *g* for 10 min. Fifty ml of ruminal fluid sample was collected by using vacuum pump linked to a stomach tube at hour 0 and 4 h post feeding. Ruminal fluid pH and temperature were tested with a hand machine (HANNA Instruments HI 8424 microcomputer, Singapore). Two parts of rumen fluid were collected into direct counts (5 mL) of bacteria and protozoal populations using Galyean’s [23] methods. Forty-five ml of ruminal fluid was added to 5 mL of 1MH_2_SO_4_ and centrifuged at 16,000× *g* for 15 min, and the supernatant was measured for NH_3_-N content [13] and underwent volatile fatty acids (VFA) analysis using HPLC [24].

### 2.4. Calculations and Statistical Analysis

For statistical analyses of the data, ANOVA was used for the 2 × 2 factorial in a 4 × 4 Latin square design using the general linear model (GLM) procedures. Statistical analyses of all replicated data were performed using the mixed procedure of SAS (2019). Treatment effects were tested using Duncan’s new multiple range test and evaluated using the method of least significant difference at the 5% significance level (*p* < 0.05).

## 3. Results and Discussion

### 3.1. Chemical Contents in the Diets

Table 1 shows the chemical content of the diets in experiment. Fermented total mixed ration contained CP, NDF, and ADF at 9.6% to 13.1.0% DM, 38.0% to 38.1.0% DM, and 24.1.0% to 24.2.0% DM, respectively. Fermented total mixed ration supplemented with 1.25% urea had a lower pH compared to that supplemented with 2.5% urea. Because ensiling time can provide soluble urea to ammonia-N, it was in the suitable range of 4.91–5.38 in FTMR. The pH affected the extent of preservation allowing nutrient content to be maintained at a high level for long periods of time. The HCN concentrations reduced 99.3% to 99.4% compared with fresh cassava root when FTMR was supplemented with 1.0% and 2.0% sulfur, respectively. Thus, HCN levels in FTMR were not dangerous for feeding animal. These levels of HCN concentration in FTMR are non-toxic for dairy cows. Larson [4] indicated that HCN concentrations for cattle over 600 ppm are toxic, and at 1000 ppm, death is usually caused. This could be due to ensiling time that can provide high temperature and acidic conditions, thus detoxifying the HCN in FTMR [25].

In addition, Kimaryo and Massawe [26] found that during the fermentation process, microorganisms can also interact directly with inactive laminarase enzymes and convert toxins into organic acids. It has been reported that when fresh cassava root was ensiled for 5 days, the HCN decreased from 176.3 to 8.2 ppm. Boonnop et al. [27] tested different fresh cassava products and showed that in the fermented product HCN reduced by 99.0% after 6 days of fermentation. The mechanism for ensiling microorganism utilizes glucose in organic acids that causes the pH to drop. Laminarase enzymes can break down activity at a lower pH to degrade linamarine to HCN, thereby reducing toxic effects [28]. In agreement with the present study, Supapong et al. [12] found that beef cattle fed a diet containing 1.0% to 2.0% sulfur experienced reduced HCN concentrations by 99.3% to 99.5%.

### 3.2. Nutrient Intake and Digestibility

The DM intake and feed digestion of animals that consumed FTMR are presented in Table 2. The total FTMR intakes were different ranging from 118.9 to 139.1 g/kg BW^0.75^. McSweeney and Denman [29] reported sulfate-supplemented animals had significantly higher (29.6%) DM intake compared with the unsupplemented group, which might have influenced by increasing in the number of rumen microbes. This is likely due to the relationship of urea and fresh cassava root, which are highly fermentable in the diet. Cows are fed ad libitum, resulting in straight fermentation and sufficient nitrogen to promote fermentation in rumen [30]. In addition, intake of CP was increased based on urea level addition (*p* < 0.05). This could provide sufficient substrate for improved digestibility coefficients and lead essential amino acids to microbial protein synthesis [7]. There was no difference in estimated energy intake between sulfur levels and urea levels on all parameters. The digestion of DM, OM, and CP when 2.0% sulfur and 2.5% urea was added to the FTMR was significantly higher (*p* < 0.05). This might be because increasing concentrations of dietary sulfur and urea may also impact rumen motility and bacterial digestibility of the diet (Table 3). Promkot et al. [31] indicated that cows exhibited increased fiber digestibility when fed a diet with 0.4% over those with 0.2% sulfur in their diet. Furthermore, Cherdthong et al. [2] found that cattle consuming fresh cassava root in the basal diet with feed block at 4.0% sulfur levels showed improved DM and OM digestibility by 13.0% and 12.1%, respectively. An earlier report by Supapong et al. [12], showed that beef cattle fed a diet supplemented with 2.0% sulfur in the FTMR exhibited increased DM digestibility by 4.2% compared to those fed with a diet supplemented with 1.0% sulfur.

### 3.3. Rumen Characteristics and Blood Profiles

Rumen parameters, ruminal pH and temperature of the FTMR groups, ranged from 6.5 to 6.6 and 39.0 to 39.4 °C, respectively (Table 3). It is well recognized to be in the range considered suitable for bacteria breakdown of fiber and activity to ferment the feed in the rumen [32]. There was no interactions between levels of sulfur and urea on the NH_3_-N concentration (*p* > 0.05), whereas ruminal NH_3_-N concentration was higher in FTMR supplemented with 2.5% urea levels (*p* < 0.01). Because urea results in the rapid degradation to ammonia by urease activity, which is excreted from rumen microbes. Ammonia-nitrogen in the rumen can occur at a much faster rate depending greatly on the availability of energy and protein in the rumen. The balance in the overall daily ratio of rumen with available carbohydrate and nitrogen in the diet may improve microbial protein production and utilization [33]. Inclusion of a high level of urea with the supplement of an extremely fermentable carbohydrate source has improved dairy cows’ milk production [34].

Blood urea-nitrogen was higher when FTMR was supplemented with 2.5% urea 4 h after morning feeding (*p* < 0.05), indicating that after feeding, effective urea may be degraded by bacteria into NH_3_-N and absorbed into the blood. In the present experiment, BUN concentration in dairy cows was not affected by sulfur (*p* > 0.05) but was always within the normal range for protein utilization. Blood thiocyanate concentration was increased by 21.6% when sulfur was supplemented at 2.0% compared to 1.0% (*p* < 0.05). National Research Council (NRC) [35] reported that the ruminant diets high in HCN level need sulfur supplementation to activate sulfur for expulsion of HCN into the thiocyanate form by rhodanese and β-mercaptopyruvate-sulfur transferase [25]. This finding was supported by the findings of Uwituze et al. [36] in sheep that were fed a cassava-based diet and exhibited a blood thiocyanate concentration directly proportional to the level of sulfur supplementation. In addition, Promkot et al. [31] added 0.5% sulfur to DM with fresh cassava foliage for beef cattle, and the findings suggested improved HCN detoxification.

There was no difference in protozoal concentration, whereas bacterial populations at 4 h after feeding were significantly greater by 6.1% with the FTMR supplemented with 2.0% sulfur and 2.5% urea (*p* < 0.01) (Table 3). In ruminants, ruminal microbes change sulphate to hydrogen sulphite, which is supplied to produce methionine and cysteine to optimize cell synthesis and maintenance for microbial synthesis. Sulfur is mainly needed to maintain maximal rumen microbial growth. Thus, the microbial population might have increased further with consistent availability of nitrogen and sulfur for fermentation in the rumen. This relationship has been investigated in in vitro studies comparing the effect of various sources of sulfur on the bacterial synthesis, the utilization of NH_3_-N, and nutrient digestion over a 96-h incubation period [37]. It can also be noted that sulfur addition increases the performance of microbial protein synthesis to production of MCP and improves the amino acid balance [38].

### 3.4. Nitrogen Balance and Purine Derivatives

The data for nitrogen balance and purine derivatives in animals treated with the FTMR are shown in Table 4. There were differences in nitrogen intake and nitrogen absorption between the two levels of urea in the FTMR (*p* < 0.05). The total nitrogen intake was highest in 2.5% urea treatments for all dairy cows. One possible explanation for this is the lack of difference in total DM intake, which showed a difference in total nitrogen intake. Allantoin concentrations, excretion, absorption, and MCP showed significant interactions between sulfur levels and urea levels in cows fed diets supplemented with 2.0% sulfur and 2.5% urea (*p* < 0.05). This may be because with urea and sulfur, there has been an apparent improvement in microbial protein synthesis associated with carbohydrate from fresh cassava root, as well as digestibility of the diet. Synchronizing the rate of supply of protein and energy sources to ruminal microbes has been conducted to maximize the capture of rumen degradable protein and to optimize efficiency of microbial protein synthesis. Both energy and protein sources should be continuously used to improve microorganism synthesis and enhance feed efficiency [39]. The microbes in rumen can benefit inorganic forms of sulfur, cooperating with NH_4_-N and carbon skeleton to form amino acids and microbial proteins. Another measure of the effect of sulfur on the activity of ruminant microorganism is the ability to synthesize protein from urea.

### 3.5. Volatile Fatty Acid (VFA) Concentration in the Rumen

The molar ratios of the VFA profile were affected by dietary FTMR (*p* < 0.01) (Table 5). Mean values of total VFA, acetic acid, propionic acid, and butyric acid were 107.1 to 120.1 m*M*, 60.2 to 64.9, 24.5 to 28.1, and 10.5 to 11.7 mol/100 mol, respectively. Furthermore, these values correlated with the nutrient digestibility when 2.0% sulfur and 2.5% urea was supplemented. Volatile fatty acid production interacted with sulfur and urea level (*p * < 0.01). Volatile fatty acids produced in the rumen are utilized by the animal as a main source of energy, and their increased production in the rumen is a measure of an increase in the efficiency of feed utilization by the animal. Sulfur in the rumen can be absorbed as sulphide (S^2−^) but also by outflow as undegraded protein sulfur or bacterial protein. Portion of degradable sulfur depends on such factors as supply of degradable nitrogen, rate of sulfur degradation by rumen microbes, and the ratio of arrival of readily fermentable energy, which affects ruminal pH and hence S^2−^ absorption [40]. Greater degradability might be due to the better activity of microbial stimulate efficiently hydrolyzed to NH_3_ by urea. This shows that the NH_3_ increased nutrient imbalances for ruminal microorganisms by enhancing the availability of carbohydrate from fresh cassava root and its ability to change into VFA production.

Different microorganisms use elemental sulfur as an electron acceptor; therefore, the reduced forms of sulfur (sulfite and sulfide) are a metabolic end product of fermentation from these microorganisms [41]. This could be because the non-protein nitrogen utilization is increased by high levels of sulfur in the rumen fluid, indicating microbial growth [5]. Furthermore, propionic acid increased by 4.6% when diets were supplemented by 2.5% sulfur (*p* < 0.01). Animals supplemented with a high amount of sulfur exhibited increased ruminal propionate concentration because propionic acid could be supplied as a sink of hydrogen sulfide when increasing ruminal available sulfur is offered [42]. Similarly, Supapong and Cherdthong [37] reported that propionic acid increased by 10.9% after supplementation of 2.0% sulfur compared to an absence of sulfur in TMR containing fresh cassava root. In addition, Promkot et al. [31] studied 1.0% sulfur supplementation in fresh cassava foliage and noted that the concentration of propionate and microbial protein synthesis were enhanced in the rumen of sulfur-supplemented cattle.

### 3.6. Milk Production, Composition, Somatic Cells and Thiocyanate Concentration

Improvement in milk yield was not significant when dairy cows were fed fresh cassava root at 40% in an FTMR diet (*p* > 0.05) containing sulfur and urea (Table 6). The milk yield of cows when the level of supplementation with sulfur and urea was not significant ranged from 12.4 to 13.0 kg/day (*p* > 0.05). However, these numerical differences could be linked to the numerical feed intake differences shown in Table 2.

Milk fat and total solids increased when feed was supplemented with 2.0% sulfur and 2.5% urea (*p* < 0.05). Efficiency of nutrient digestibility increased when absorbed as glucose rather than when nutrients utilized by rumen microbes and the propionate converted to glucose in the liver [43]. This enhances the proportion of amino acids and glucose relative to that of acetate and long chain fatty acids in the circulation, resulting in enhanced production of protein, lactose, and to a lesser degree, fat in the mammary gland [44,45,46]. Knika and Zmiev [45] observed that supplementation of dairy cow rations with 30 g of sodium sulphate per day for 30 days increased cellulose digestibility by 13%. Dageaw et al. [46] found that as a result of the sulfur treatment, the production of milk solids, fat, protein, and casein increased in dairy cows fed a diet supplemented with 1.5% BW fresh cassava root combined with 4.0% feed block containing high sulfur and increased milk fat by 8.6%. This is consistent with the result that milk fat increases by 3.7% when the diet is supplemented with 2.0% sulfur and 2.5% urea. High propionic acid using the substrate to synthesize glucose in gluconeogenesis by the pentose phosphate pathway produces NADPH, which is a factor in the synthesis of fatty acids [47,48,49]. Feed high in NDF is related to an improved production rate of lipogenic to glucogenic VFA, with the alteration in the proportion of VFA resulting in an enhanced concentration of milk fat [1,50,51,52].

In this study, the HCN result was critical differentiation in the milk thiocyanate concentration which is related to the levels of sulfur fortified in feed (Table 6). The diets supplemented with 2.0% sulfur levels resulted in greater concentrations of milk thiocyanate (*p* < 0.05); however, the result was not significantly different based on urea levels (*p* > 0.05). In the present data, milk thiocyanate ranged from 5.02 to 11.87 ppm. The advantage of a significant dose of HCN in fresh cassava root is to change the thiocyanate through the elimination of feed HCN by changing it to thiocyanate in the liver and kidney of cows by the rhodanese enzyme action and partly through detoxification in the milk [11], which was observed in milk thiocyanate increases. Srisaikham et al. [48] reported that effects on milk thiocyanate in the case of dairy cows were also positively correlated in a regression model between milk thiocyanate and HCN in dietary fresh cassava peel results.

The present experiment was conducted in the Research Center of Institute, which controlled external factors such as dirty environment, substrates, or health factors such as pre-clinical mastitis. None of these variables were investigated for in the study; thus, they have not impacted somatic cell count results. Thus, we could be confident that the somatic cell count results were influenced by the treatment study. The decrease in somatic cell count was possibly because the rhodanese enzyme in the liver contains sulfur and transformed HCN into the blood thiocyanate, which is transferred to the milk, saliva, and urine [11]. The lactoperoxidase system was affected in the antimicrobial function, activated by milk thiocyanate in the milk, to limit the increase in somatic cell count. This effect was due to the restraining the activity of various cytoplasmic enzymes to damage the cell bacterial membranes and reduce growth [9,51,52]. These characteristics enable milk thiocyanate and somatic cell count in the milk and can be activated by lactoperoxidase as an indicator of mastitis, which has been previously elucidated by Isobe et al. [49]. Thus, milk thiocyanate from HCN can be used to prolong the shelf life of raw milk stored at room temperature by inhibiting microbial growth [48].

## 4. Conclusions

Based on this research, it could be concluded that the inclusion of 2.0% sulfur with 2.5% urea in FTMR containing fresh cassava root improved digestibility, ruminal fermentation, and microbial crude protein synthesis in dairy cows. In addition, milk fat and total solids increased, whereas somatic cell count reduced when FTMR was supplemented with 2.0% sulfur and 2.5% urea. The HCN concentrations reduced 99.3% to 99.4% compared with fresh cassava root when FTMR was supplemented with 1.0% and 2.0% sulfur. However, this research is based on a few animals; therefore, results from this study should be interpreted cautiously. Further research should elucidate the effect of FTMR supplemented with sulfur and urea in production trials with a high number of cows and a longer period than the current study.

## Figures and Tables

**Table 1 vetsci-07-00098-t001:** Ingredient and chemical composition of fermented total mixed ration (%DM).

Item	1% Sulfur		2% Sulfur		Fresh Cassava Root
	1.25% Urea	2.5% Urea	1.25% Urea	2.5% Urea	
Ingredients, %DM					
Rice straw	40.0	40.0	40.0	40.0	
Fresh cassava root	40.0	40.0	40.0	40.0	
Soybean meal	5.0	5.0	5.0	5.0	
Palm kernel meal	4.7	3.5	3.7	2.5	
Rice bran	3.0	3.0	3.0	3.0	
Urea	1.3	2.5	1.3	2.5	
Pure sulfur	1.0	1.0	2.0	2.0	
Mineral premix	1.0	1.0	1.0	1.0	
Molasses, liquid	3.0	3.0	3.0	3.0	
Salt	1.0	1.0	1.0	1.0	
Chemical composition					
Dry matter, %	55.2	55.6	55.4	55.3	32.0
Organic matter, %DM	96.3	96.6	95.6	95.8	96.3
Ash, %DM	3.7	3.4	4.4	4.2	3.7
Crude protein, %DM	9.6	13.0	9.8	13.1	2.8
Neutral detergent fiber, %DM	38.1	38.1	38.0	38.1	7.8
Acid detergent fiber, %DM	24.2	24.2	24.1	24.2	5.7
pH	5.09	5.38	4.91	5.15	
Hydrocyanic acid, ppm	0.76	0.70	0.76	0.74	110.00

**Table 2 vetsci-07-00098-t002:** Influence of fermented total mixed ration on dry matter intake, nutrient intake and digestibility coefficients in dairy cows.

Item	1.0% Sulfur		2.0% Sulfur		**SEM**	*p*-Value		
	1.25% Urea	2.5% Urea	1.25% Urea	2.5% Urea		S	U	S*U
Dry matter intake								
% BW	2.5	2.8	2.8	3.0	0.51	0.45	0.35	0.85
g/kg BW^0.75^	118.9 ^a^	132.0 ^b^	129.3 ^b^	139.1 ^b^	3.37	0.05	0.05	0.89
Nutrient intake, kg/day								
Organic matter	12.0	13.4	12.9	13.9	1.05	0.54	0.29	0.88
Crude protein	1.2 ^a^	1.8 ^b^	1.3 ^a^	1.9 ^b^	0.34	0.41	0.01	1.00
Neutral detergent fiber	4.8	5.3	5.1	5.6	0.66	0.50	0.30	0.96
Acid detergent fiber	3.0	3.4	3.3	3.5	0.53	0.47	0.29	0.90
Estimated energy intake								
DOMI ^d^, kg/day	8.4	9.6	9.2	10.0	0.88	0.48	0.24	0.83
DOMR ^e^, kg/day	5.5	6.2	6.0	6.5	0.71	0.45	0.23	0.81
ME, MJ/day	32.0	36.4	34.8	38.0	2.72	0.47	0.23	0.85
Nutrient digestibility, %								
Dry matter	61.0 ^a^	66.3 ^b^	68.5 ^b^	69.0 ^b^	1.07	0.01	0.03	0.06
Organic matter	70.0 ^a^	71.5 ^b^	71.0 ^b^	71.8 ^b^	0.33	0.15	0.02	0.37
Crude protein	61.0 ^a^	68.3 ^b^	63.5 ^a^	69.0 ^b^	1.40	0.42	0.01	0.66
Neutral detergent fiber	59.5	59.0	59.8	60.3	1.44	0.73	1.00	0.81
Acid detergent fiber	38.0	37.3	38.5	40.3	1.25	0.29	0.76	0.44

S: *p*-value level of sulfur in diet. U: *p*-value level of urea in diet. S*U: *p*-value interaction between level of sulfur and urea in diet. SEM: standard error of mean. ^a,b^ Means within rows with different letters differ (*p* < 0.05). BW: Body weight of dairy cow. ^d^ DOMI: Digestible organic matter intake. ^e^ DOMR: Digestible organic matter fermented in the rumen. ME: Metabolisable energy.

**Table 3 vetsci-07-00098-t003:** Effects of fermented total mixed ration on rumen ecology, microorganism, blood urea-nitrogen and blood thiocyanate in dairy cows.

Item	1% Sulfur		2% Sulfur		SEM	*p*-Value		
	1.25% Urea	2.5% Urea	1.25% Urea	2.5% Urea		S	U	S*U
Rumen ecology								
Ruminal pH								
0 h post feeding	6.6	6.6	6.8	6.7	0.33	0.24	0.24	0.58
4 h post feeding	6.4	6.4	6.4	6.2	0.30	0.44	0.44	0.30
Ruminal temperature, °C								
0 h post feeding	38.9	38.9	39.2	39.4	0.56	0.21	0.72	0.60
4 h post feeding	39.2	39.3	39.6	39.4	0.35	0.09	0.84	0.32
NH_3_-N concentration, mg/dL								
0 h post feeding	8.3 ^a^	10.0 ^b^	9.3 ^a^	11.0 ^b^	0.54	0.11	0.01	0.93
4 h post feeding	16.0 ^a^	23.0 ^b^	16.2 ^a^	22.9 ^b^	0.50	0.81	0.01	0.53
Ruminal microbes, cell/mL								
Protozoa, ×10^6^								
0 h post feeding	9.5	9.5	10.0	9.0	0.45	1.00	0.43	0.33
4 h post feeding	12.5	12.8	13.3	12.5	0.67	0.58	0.58	0.28
Bacteria, ×10^9^								
0 h post feeding	30.8	31.8	31.0	32.0	0.70	0.62	0.06	1.00
4 h post feeding	41.3 ^a^	42.3 ^a^	42.0 ^a^	43.8 ^b^	0.48	0.03	0.01	0.43
Blood metabolites								
Blood urea-N concentration, mg/dL								
0 h post feeding	10.3	12.0	10.8	13.0	1.38	0.70	0.32	0.90
4 h post feeding	11.0 ^a^	14.0 ^bc^	12.3 ^ab^	15.0 ^c^	1.03	0.31	0.02	0.91
Blood thiocyanate concentration, mg/dL								
0 h post feeding	12.6	12.7	15.0	14.6	1.21	0.17	0.91	0.87
4 h post feeding	12.5 ^a^	14.0 ^ab^	15.8 ^bc^	17.6 ^c^	0.97	0.01	0.12	0.85

S: *p*-value level of sulfur in diet. U: *p*-value level of urea in diet. S*U: *p*-value interaction between level of sulfur and urea in diet. SEM: standard error of mean. ^a,b,c^ Means within rows with different letters differ (*p* < 0.05).

**Table 4 vetsci-07-00098-t004:** Effect of fermented total mixed ration supplementation on nitrogen (N) balance and purine derivative.

Item	1.0% Sulfur		2.0% Sulfur		SEM	*p*-Value		
	1.25% Urea	2.5% Urea	1.25% Urea	2.5% Urea		S	U	S*U
N balance, g/day								
N intake	209.1 ^a^	230.6 ^b^	225.3 ^b^	243.7 ^b^	4.35	0.45	0.05	0.94
N excretion								
Feces	82.5	77.1	78.5	80.5	3.07	0.98	0.86	0.70
N balance								
Absorption	126.6 ^a^	153.4 ^b^	146.9 ^b^	163.1 ^c^	3.46	0.24	0.05	0.67
Purine derivative, mmol/day								
Allantoin, mmol/day								
Excretion	194.6 ^a^	218.8 ^c^	203.8 ^b^	243.6 ^c^	1.66	0.01	0.01	0.02
Absorption	195.8 ^a^	224.2 ^c^	206.7 ^b^	253.4 ^c^	1.80	0.01	0.01	0.02
Creatinine, mg/dL	26.9 ^a^	26.1 ^ab^	26.1 ^ab^	25.4 ^b^	0.48	0.01	0.01	0.83
MCP, g/day	664.5 ^a^	747.2 ^c^	696.2 ^b^	831.8 ^c^	3.07	0.01	0.01	0.02
MCP (g/digest OM kg)	74.1	78.7	76.8	86.1	4.86	0.20	0.13	0.34
EMNS (gN/kg OMDR)	11.1 ^a^	11.1 ^a^	13.1 ^a^	14.9 ^b^	1.03	0.02	0.46	0.42

S: *p*-value level of sulfur in diet. E: *p*-value level of urea in diet. S*U: *p*-value interaction between level of sulfur and urea in diet. SEM: standard error of mean. ^a,b,c^ Means within rows with different letters differ (*p* < 0.05). OM: organic matter. MCP: microbial crude protein. Efficiency of microbial N synthesis (EMNS, g/kg of OM digested in the rumen (OMDR) = [(MCP (g/day) × 1000)/DOMR (g)], assuming that rumen digestion was 650 g/kg OM of digestion in total tract (ARC, 1990).

**Table 5 vetsci-07-00098-t005:** Volatile fatty acid (VFA) of rumen fluid of dairy cow fed fermented total mixed ration.

Item	1.0% Sulfur		2.0% Sulfur		SEM	*p*-Value		
	1.25% Urea	2.5% Urea	1.25% Urea	2.5% Urea		S	U	S*U
Total VFA, m*M*								
0 h post feeding	106.0	107.5	107.1	106.9	0.65	0.58	0.17	0.06
4 h post feeding	108.1 ^a^	129.4 ^bc^	126.6 ^b^	133.3 ^c^	1.21	0.01	0.01	0.01
VFA profiles, mol/100 mol								
Acetic acid								
0 h post feeding	64.8	64.0	64.8	64.2	0.69	0.78	0.16	0.86
4 h post feeding	65.0 ^a^	58.9 ^bc^	60.8 ^b^	56.2 ^c^	0.83	0.01	0.01	0.01
Propionic acid								
0 h post feeding	24.0 ^a^	26.1 ^b^	24.1 ^a^	26.4 ^b^	0.65	0.71	0.05	0.84
4 h post feeding	25.0 ^a^	29.6 ^b^	29.3 ^b^	29.9 ^b^	0.58	0.01	0.01	0.01
Butyric acid								
0 h post feeding	11.2 ^a^	9.0 ^ab^	10.1 ^ab^	8.5 ^b^	0.78	0.63	0.03	0.78
4 h post feeding	10.0 ^a^	12.5 ^bc^	10.9 ^ab^	14.9 ^c^	0.85	0.04	0.01	0.32

S: *p*-value level of sulfur in diet. E: *p*-value level of urea in diet. S*U: *p*-value interaction between level of sulfur and urea in diet. SEM: standard error of mean. SEM: standard error of mean. ^a,b,c^ Means within rows with different letters differ (*p* < 0.05). VFA: Volatile fatty acid.

**Table 6 vetsci-07-00098-t006:** Effect of fermented total mixed ration on milk production and chemical composition.

Item	1% Sulfur		2% Sulfur		SEM	*p*-Value		
	1.25% Urea	2.5% Urea	1.25% Urea	2.5% Urea		S	U	S*U
Milk production								
Milk yield, kg/day	12.4	12.5	12.8	13.0	0.83	0.55	0.81	0.92
3.5% FCM, kg/day	13.0	13.4	13.6	13.9	1.01	0.61	0.75	0.96
Milk composition, %								
Fat	3.55 ^a^	3.63 ^ab^	3.66 ^b^	3.68 ^b^	0.03	0.01	0.01	0.22
Protein	3.22	3.44	3.33	3.50	0.13	0.15	0.07	0.25
Lactose	4.29	4.34	4.36	4.41	0.11	0.55	0.66	1.00
Solids-not-fat	8.21	8.48	8.39	8.61	0.15	0.11	0.07	0.30
Total solids	11.76 ^a^	12.11 ^a^	12.05 ^a^	12.29 ^b^	0.16	0.06	0.04	0.25
Milk thiocyanate, ppm	5.02 ^a^	6.02 ^a^	11.87 ^b^	10.48 ^b^	1.42	0.02	0.92	0.56
Somatic cell count, cell/mL	262,833 ^a^	287,833 ^a^	226,500 ^b^	223,541 ^b^	146.95	0.04	0.62	0.53

S: *p*-value level of sulfur in diet. U: *p*-value level of urea in diet. S*U: *p*-value interaction between level of sulfur and urea in diet. FCM: fat collected milk. SEM: standard error of mean. SEM: standard error of mean. ^a,b^ Means within rows with different letters differ (*p* < 0.05).

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
