# Peer review of "Effect of Sulfur and Urea Fortification of Fresh Cassava Root in Fermented Total Mixed Ration on the Improvement Milk Quality of Tropical Lactating Cows"

_vetsci, 2020, doi:10.3390/vetsci7030098_

Round 1

Reviewer 1 Report

General comments:

In summary, this research investigated the effect of fresh cassava root in fermented total mixed ration with sulfur and urea on milk quality of tropical lactating dairy cows.

I believe this research is interesting as it utilizes a feed source commonly available in tropical zones which, if it improved animal performance without any negative side effects on health or otherwise, could be a practical and viable feed additive solution.

My main concern with this study is from an animal welfare concern standpoint, I hope that a clarification will ensure that these cows were not housed individually throughout the trial. Individual housing of cows is an extremely stressful procedure for the cow and should be avoided for any extended period of time.

In addition, I do have questions regarding the few animals used for this trial in combination with a general lack of detail regarding animals used, statistics, animal housing, environment, health and importantly, feeding regime. If these areas could be expanded and clarified that would benefit the manuscript greatly.

See line by line comments below.

Introduction

Ln 43: In ruminant(s).

Ln 48-50. Consider reworking these two sentences slightly for clarity.

Ln 52-54: Is the of the sentence supposed to say “…,which has efficient antimicrobial properties.”? Please revise or clarify.

Ln 62-63: This sentence needs clarification. Were there two hypotheses or only one? Currently it reads like one hypothesis is missing. Please advise.

Material and methods and Calculations and statistical analysis

This section needs work and clarification.

Was there any missing data? When was the study conducted? Could there have been seasonal effects (temperature, humidity etc) affecting the data? Based on the location, temperature and mitigation, heat stress could have impacted rumination behavior and possibly digestibility of the experimental feeds.

How were these cows housed?  It states that cows were fed in individual pens, but does this imply individual housing throughout the study or only at feeding? Were these cows housed in a free-stall environment, tie-stall or bedded pack system? What type of bedding was provided? How often was it replaced/replenished? Were all cows healthy throughout the study? This could have had an impact on your somatic cell count and other variables.

Also, depending on the setup, prolonged individual housing is also highly stressful and could have confounded the study by altering the baseline as well as obtained data from the trial period.

How was the feed intake recorded? Were automatic feeders used or did you manually weigh orts post-feeding? Please clarify how the cows were fed, it is currently confusing for how long cows had access to feed. Was the feed sitting out? What impact might this have had based on when cows consumer the feed? These are important details left out for a nutritional study.

The mineral block was provided by free choice, did all cows end up taking advantage of the mineral block? If not, could the composition of the mineral block have had an impact on the rumen fluid samples for cows that did or did not use the mineral block?

Was this study conducted on 4 groups of cows, or 4 cows total as in (n=4)? If the previous, the total starting number of cows for each group should be provided including the number of cows that completed the trial.

If the latter (which I’m assuming is the case), I have difficulty seeing how the small number of 4 cows used would not have a major interpretational impact on the data due to the possibility of individual (or genetical) variability. Even if the power analysis across the factorial and LS designs may be sound with only 4 cows; the biological relevance, in regard to the possibility of individual (genetic) variability, that could possibly have influenced the results and conclusions made thereof. To be certain that the reported effects are sounds, additional cows or replicates would have been beneficial.

Was a power analysis made or was this design based on another study? What was the rationale for only using 4 cows if that was the case? In essence, if I understand this correctly, you have 16 observations but are comparing treatment groups of n=4, thus the treatments sample sizes would have benefitted from more cows in the trial. This is a very low sample size. I'd suggest that the DF and F values should be added where applicable from the analyses made.

Results and Discussion

Ln 154:  add “which are” between “…root,” and “highly”.

Ln 155-156: Which ruminants? This statement is a generalization. Not all ruminants are fed ad libitum. In your study you also state that the cows were fed ad libitum but only in the morning and in the afternoon. This is confusing. Did the cows have free access between 07-16:00 with fresh feed provided at 16:00 or did they only have access to feed during a specific time frame? Please clarify.

Ln 162-163: This statement is referring to data from the cited work but may be confusing as it sounds as it is data related to your study. Consider rewording.

Ln 213-217: Again, how was feed intake measured, and how accurately was this data obtained. There are clearly numerical differences in Table 2. If this data was only estimated, there is plenty of room for potential errors.

Ln 271-273: These numerical differences could also be linked to the numerical feed intake differences shown in table 2.

Ln 304-315: This entire section is discussed from a standpoint of physiology and biochemistry without taking external factors into account, such as a dirty environment, substrates, or health, such as pre-clinical mastitis. Were any of these variables investigated or controlled for in the study, and could they have impacted your SCC results?

Conclusions and recommendations

Since this research is based on few animals, this should be acknowledged as a limitation in the conclusions and that results from this study should be interpreted cautiously.

References

Please look over the reference list for missing and any mis-numbered citations. Maybe its only a sorting issue currently as references 2,3,4,5 and 6are not in order, but there might be errors as well.

Author Response

We highly appreciated all the comments and suggestions made by the two reviewers. Above all, the authors felt that all points made were very useful and have incorporated most of the corrections where necessary as suggested in order to make the manuscript ready for possible publication. All those corrected and modified appear in "Track Changes" function in Microsoft Word of the manuscript. Please see information given by the authors following the suggestions and comments made by the three reviewers. 

With the above information we would like to resubmit our paper for your kind considerations for a possible publication in Veterinary Science. 

We again wish to thank you very much for your kind attention and support. 

Sincerely yours, 

Assoc. Prof. Dr. Anusorn Cherdthong 

Contact: [email protected] 

Response to Reviewer 1: 

General comments: 

In summary, this research investigated the effect of fresh cassava root in fermented total mixed ration with sulfur and urea on milk quality of tropical lactating dairy cows. 

I believe this research is interesting as it utilizes a feed source commonly available in tropical zones which, if it improved animal performance without any negative side effects on health or otherwise, could be a practical and viable feed additive solution. 

My main concern with this study is from an animal welfare concern standpoint, I hope that a clarification will ensure that these cows were not housed individually throughout the trial. Individual housing of cows is an extremely stressful procedure for the cow and should be avoided for any extended period of time. 

In addition, I do have questions regarding the few animals used for this trial in combination with a general lack of detail regarding animals used, statistics, animal housing, environment, health and importantly, feeding regime. If these areas could be expanded and clarified that would benefit the manuscript greatly. 

Response: We highly appreciate thanks to Reviewer 1, who have provided a great positive comment on our manuscript. We hope that if our work could be considered for publication, it will more useful to the farmer in the tropical zone who have grown cassava. Based on this experiment (84 das), we found that the negative side did not occur to animals, whereas improved animal performance when feeding with dietary treatment. In addition, we know that the individual pen (5x5 m2of cows might affect animal stress; thus, before starting the experiment, the animals must training-in with an individual pen for at least 2 weeks. After that, animals can adapt well to individual pens and can start an experiment. The objective of an individual pen is that individual feed intake can be recorded and easy to manage. Lastly, individual pen animals are the international standard methodology for evaluating feed digestion trials which is acceptable for ruminant nutritionists across the country. Moreover, there are several international high standard journals are accept similar methodology and also a few animals published in their Journal (for example: Seankamsorn and Cherdthong (2020) Animals. 10 (1): 56, Cherdthong et al. (2019) Animals. 9(4): 130; Cherdthong et al. (2018) Journal of Animal Physiology and Animal Nutrition. 102(6):1509-1514). For more details, we have provided in each comment. Thus, we hope that the Reviewer 1 could reconsider our reason.   

See line by line comments below. 

Introduction 

Ln 43: In ruminant(s). 

Response: We have revised as suggestion. Please see in manuscript Line 43 

Ln 48-50. Consider reworking these two sentences slightly for clarity. 

Response: We have revised as suggestion as “Supplementing non-protein nitrogen to animal diets is to provide ammonia-nitrogen (NH3-N) that might be incorporated in their proteins by the rumen microorganism. Due to the fact that microbial rumen needed supplemental sulfur when feeding non-protein nitrogen to completely utilize NH3-N for protein synthesis [1,6].” We have revised as suggestion. 

Ln 52-54: Is the of the sentence supposed to say “…,which has efficient antimicrobial properties.”? Please revise or clarify. 

Response: We have revised as suggestion. Please see in manuscript Line 55-56 

Ln 62-63: This sentence needs clarification. Were there two hypotheses or only one? Currently it reads like one hypothesis is missing. Please advise. 

Response: It consisted one hypothesis and we have modified as “It was hypothesized that FTMR containing of fresh cassava root with sulfur and urea addition for ruminant diets and can also improve feed efficiency and milk quality in cows” Please see in Line 64 

Material and methods and Calculations and statistical analysis 

This section needs work and clarification. 

Was there any missing data? When was the study conducted? Could there have been seasonal effects (temperature, humidity etc) affecting the data? Based on the location, temperature and mitigation, heat stress could have impacted rumination behavior and possibly digestibility of the experimental feeds. 

Response: Thank you for your concern. There was no missing data. This experiment was done during October 2018 - January 2019, which is a winter season in Thailand, so the temperature (25-32°C) did not much effect on animal stress. Cows were housing in the research station of the institute, which is specific to arrange to do the research. Thus, based on our facilitate, rumination behavior and digestibility were influenced by the environmental effect. 

How were these cows housed?  It states that cows were fed in individual pens, but does this imply individual housing throughout the study or only at feeding? Were these cows housed in a free-stall environment, tie-stall or bedded pack system? What type of bedding was provided? How often was it replaced/replenished? Were all cows healthy throughout the study? This could have had an impact on your somatic cell count and other variables. 

Response: As we mentioned above, cows were individual pen (5x5 m2) in a free-stall environment throughout the study. Rubber mats were provided as bedding (replaced every 21 days) and clean every morning. Animals were healthy throughout the study. Thus, we confident that no environmental effect occurs to study. 

Also, depending on the setup, prolonged individual housing is also highly stressful and could have confounded the study by altering the baseline as well as obtained data from the trial period. 

Response: We know that the individual pen (5x5 m2of cows might affect animal stress; thus, before starting the experiment, the animals must training-in with an individual pen for at least 2 weeks. After that, animals can adapt well to individual pens and can start an experiment. 

How was the feed intake recorded? Were automatic feeders used or did you manually weigh orts post-feeding? Please clarify how the cows were fed, it is currently confusing for how long cows had access to feed. Was the feed sitting out? What impact might this have had based on when cows consumer the feed? These are important details left out for a nutritional study. 

Response: Feed intake can be recorded cow by cow at morning (07:00) and afternoon (16:00)] by manually weigh feed offered (today) and residues (next day before new feeding time). Sitting out was well prevented by house design and in case remain sitting out will collect and weighed. Thus, methodology of feed intake record was done very careful in order to left out for a nutritional study. 

The mineral block was provided by free choice, did all cows end up taking advantage of the mineral block? If not, could the composition of the mineral block have had an impact on the rumen fluid samples for cows that did or did not use the mineral block? 

Response: Sorry, mineral block did not provided. So, we have modified in the manuscript. 

Was this study conducted on 4 groups of cows, or 4 cows total as in (n=4)? If the previous, the total starting number of cows for each group should be provided including the number of cows that completed the trial. 

Response: 4 cows total as in (n=4) was used in a Latin Square Design, which we have indicated in section of 2.1. 

If the latter (which I’m assuming is the case), I have difficulty seeing how the small number of 4 cows used would not have a major interpretational impact on the data due to the possibility of individual (or genetical) variability. Even if the power analysis across the factorial and LS designs may be sound with only 4 cows; the biological relevance, in regard to the possibility of individual (genetic) variability, that could possibly have influenced the results and conclusions made thereof. To be certain that the reported effects are sounds, additional cows or replicates would have been beneficial. 

Response: We have agreed with your comments. However, limitation of our research station is few number cows with the same condition (DIM, BW, lactation etc.), thus, we cannot supply more cows into experiment. Lowering an error by man, sample collection and animal management were carefully done. As above limitation, there are several international high standard journals still accepted to publish in their Journal. 

Was a power analysis made or was this design based on another study? What was the rationale for only using 4 cows if that was the case? In essence, if I understand this correctly, you have 16 observations but are comparing treatment groups of n=4, thus the treatments sample sizes would have benefitted from more cows in the trial. This is a very low sample size. I'd suggest that the DF and F values should be added where applicable from the analyses made. 

Response: You are corrected. There are 4 cows and 16 observations were obtained by LS design. Even a low sample size but with 4 x 4 Latin Square is acceptable for publication by various journals (for example: Seankamsorn and Cherdthong (2020) Animals. 10 (1): 56, Cherdthong et al. (2019) Animals. 9(4): 130; Cherdthong et al. (2018) Journal of Animal Physiology and Animal Nutrition. 102(6):1509-1514).  

Results and Discussion 

Ln 154:  add “which are” between “…root,” and “highly”. 

Response: We have revised as suggestion. Please see in manuscript Line 154 

Ln 155-156: Which ruminants? This statement is a generalization. Not all ruminants are fed ad libitum. In your study you also state that the cows were fed ad libitum but only in the morning and in the afternoon. This is confusing. Did the cows have free access between 07-16:00 with fresh feed provided at 16:00 or did they only have access to feed during a specific time frame? Please clarify. 

Response: We changed to “Cows”. In addition, cows have free access between 07-16:00 with fresh feed provided at 16:00. 

Ln 162-163: This statement is referring to data from the cited work but may be confusing as it sounds as it is data related to your study. Consider rewording. 

Response: Thanks, we have modified as “Promkot et al. [31] indicated that cows exhibited increased fiber digestibility when fed a diet with 0.4% over those with 0.2% sulfur in their diet.” Please see in manuscript Line 162-163 

Ln 213-217: Again, how was feed intake measured, and how accurately was this data obtained. There are clearly numerical differences in Table 2. If this data was only estimated, there is plenty of room for potential errors. 

Response: Feed intake was measured as above mentioned.  

Ln 271-273: These numerical differences could also be linked to the numerical feed intake differences shown in table 2. 

Response: We have added as suggestion as “Milk yield was not significant when dairy cows were fed fresh cassava root at 40% in an FTMR diet (p > 0.05) containing sulfur and urea (Table 6). The milk yield of cows when the level of supplementation with sulfur and urea was not significant ranged from 12.4 to 13.0 kg/d (p > 0.05). However, these numerical differences could be linked to the numerical feed intake differences shown in Table 2. This yield was in the optimal range of tropical zone, which is typically associated with animal feeds and arrangement, animal genetics, and environment. This may also indicate the lack of significance compared with feed consumed among the animals; hence, no notable difference emerged for milk yield between the treatments.” Please see in Line 268 

Ln 304-315: This entire section is discussed from a standpoint of physiology and biochemistry without taking external factors into account, such as a dirty environment, substrates, or health, such as pre-clinical mastitis. Were any of these variables investigated or controlled for in the study, and could they have impacted your SCC results? 

Response: As above mentioned, this experiment was done in Research Center of Institute, which is control external factors such as dirty environment, substrates, or health, such as pre-clinical mastitis. Thus, it was confident that the results influenced by treatment study. 

Conclusions and recommendations 

Since this research is based on few animals, this should be acknowledged as a limitation in the conclusions and that results from this study should be interpreted cautiously. 

Response: Thank you. Now we have added as “….However, since this research is based on a few animals; therefore, results from this study should be interpreted cautiously…. Please see in section of conclusion. 

References 

Please look over the reference list for missing and any mis-numbered citations. Maybe its only a sorting issue currently as references 2,3,4,5 and 6are not in order, but there might be errors as well. 

Response: Thanks so much and now we have modified. Please see in Reference list. 

Response to Reviewer 2: 

Reviewer comments for manuscript ID vetsci-855751 entitled ‘Fresh Cassava Root in Fermented Total Mixed Ration containing Sulfur and Urea Levels could Improve Milk Quality of Tropical Lactating Cows’ 

General comments 

The authors have touched a growing field of animal nutrition research using locally available materials and their fortification with organic/inorganic compounds to improve animal productivity and performance. This research should be able to provide alternatives to traditional dairy feeding through cost cutting in production. I congratulate the authors for this work. I have some comments on the manuscript that I would like the authors to ponder and do the needful before recommendation for publication. 

Response: The authors would like to sincere thanks to the Reviewer 2 who provided a positive comments and appreciate in our work. All comment and suggestion have been all agreed and revised accordingly point-by-point. Thus, we believed that this work would be useful to the readers. Thanks! 

Introduction: Please elaborate on the gaps in literature you have identified and aim to fill through this research. 

Response: Thank you, we have tried our best to modify throughout manuscript. We hope that it will met to your suggestion. 

Results and discussion: This is overall a well written portion especially the discussion. However care must be taken at some points that I have specifically pointed out to properly clarify the points of discussion. 

Response: All comment and suggestion have been all agreed and revised accordingly point-by-point. 

Conclusions and recommendations: Conclusions are too short and do not elaborate about the limitations of this research and future directions of the research. Polioencephalaomalacia due to sulphur supplementation as a passing reference in the end might put doubts in the applicability of this research. This point must be carefully dealt and elaborated. I found no recommendations. 

Response: Thank you for suggestion. Now, we have modified as “Based on this research it could be concluded that inclusion of 2.0% sulfur with 2.5% urea in FTMR containing fresh cassava root improved digestibility, ruminal fermentation, and microbial crude protein synthesis in dairy cows. In addition, milk fat and total solids increased, whereas somatic cell count reduced when FTMR was supplemented with 2.0% sulfur and 2.5% urea. The HCN concentrations reduced 99.3% to 99.4% compared with fresh cassava root when FTMR was supplemented with 1.0% and 2.0% sulfur. However, since this research is based on a few animals; therefore, results from this study should be interpreted cautiously.  Further research should elucidate the effect of FTMR supplemented with sulfur and urea in production trials with a high number of cows and a longer period than the current study. In addition, some modification have been done according to the comments by Reviewer 1. 

Specific comments 

Line 48: I could not understand this line. Please clarify ‘Non-protein nitrogen is used to analyze microbes in rumen because sulfur became limited. ‘ 

Response: We have modified as “Supplementing non-protein nitrogen to animal diets is to provide ammonia-nitrogen (NH3-N) that might be incorporated in their proteins by the rumen microorganism.” Please see in manuscript Line 48 

Line50-52: Please reframe ‘Sulfur and nitrogen are combined to improve the rumen microbes to synthesize amino acids, and digestion of cellulose could be attributable to activity of fiber fermentation [7]’ as ‘Sulfur and nitrogen are combined to improve the synthesize of amino acids, and digestion of cellulose by rumen microbes through higher activity of fiber fermentation [7]’ 

Response: We have revised as suggestion. Please see in manuscript Line 51-52 

Line 54: Please clarify this sentence ‘…. which has efficient antimicrobial pathogens’ 

Response: We have modified as “…, which has efficient antimicrobial properties” Please see in manuscript Line 55 

Lines 56-57: Please reframe ‘These have catalyzed the oxidation of thiocyanate by hydrogen peroxide with the synthesis of the antibacterial hypothiocyanite and other substrates’ as ‘These catalyze the oxidation of thiocyanate by hydrogen peroxide through the synthesis of the antibacterial hypothiocyanite and other substrates’ 

Response: We have revised as suggestion. Please see in manuscript Line 58-59 

Line 59: Please replace ‘synthetic’ with ‘synthesized’ 

Response: We have revised as suggestion. Please see in manuscript Line 61 

Line 62: Please reframe the sentence ‘It was hypothesized that fresh cassava root with sulfur and urea addition for ruminant diets and can also improve animal performance and feed efficiency’ as ‘It was hypothesized that fresh cassava root with the addition of sulfur and urea in ruminant diets improves animal performance and feed efficiency’ 

Response: We have revised as “It was hypothesized that FTMR containing fresh cassava root with sulfur and urea addition improves feed efficiency, rumen fermentation efficiency, and milk quality” Please see in manuscript Line 64 

Line 68: Please replace ‘allowed’ with ‘approved’ 

Response: We have revised as suggestion. Please see in manuscript Line 70 

Lines77-79: These sentences are contradictory ‘All dairy cows were fed in individual pens on an ad libitum basis. Animals were allowed free access to feed in the morning (07:00) and afternoon (16:00), while water and mineral block were provided by free choice’ Please clarify. 

Response: We have revised as “FTMR was fed ad libitum [morning (07:00) and afternoon (16:00)] allowing for 10% refusals. All cows were kept in individual pens, and clean fresh water and mineral blocks were available at all times. ” Please see in manuscript Line 79-80 

Lines 80-81: Please reword ‘….samples, rumen fluid, feces, and blood’ as ‘…… samples of rumen fluid, feces, and blood’ 

Response: We have revised as suggestion. Please see in manuscript Line 82 

Lines 82-86: Please reframe ‘The FTMR was made according to Supapong et al. [12]. In short, the FTMR was made by packing concentrate mixed with rice straw chopped to a length of 3-5 cm by machine and adding water to achieve a moisture content of 55% in processing. FTMR was packed in plastic bags to ferment under anaerobic conditions and stored indoors (25-32 °C) for 7 days before use’ as ‘FTMR was made by packing concentrate mixed with rice straw chopped to a length of 3-5 cm by machine and adding water to achieve a moisture content of 55% in processing, as per the process carried out by Supapong et al. [12]. FTMR was then packed in plastic bags to ferment under anaerobic conditions and stored indoors (25-32 °C) for 7 days before use’ 

Response: We have revised as suggestion. Please see in manuscript Line 84-87 

Line 97: Please add concentration after ‘allantoin and creatinine’ to read as ‘….. allantoin and creatinine concentrations’ 

Response: We have revised as suggestion. Please see in manuscript Line 99 

Lines 110-112: Please rewrite the sentence again as ’50 ml of ruminal fluid sample was collected by using vacuum pump linked to a stomach tube at hour 0 and 4 h post feeding’ 

Response: We have revised as suggestion. Please see in manuscript Line 112-113 

Line115: Please rewrite ‘The ruminal fluid 45 ml was added...’ as ‘45ml of ruminal fluid was added….’ 

Response: We have revised as suggestion. Please see in manuscript Line 116 

Lines 126-27: Abbreviations used for the first time must be written in expanded form. 

Response: There are first provided in section of 2.2 already. Please see in manuscript Line 127 

Lines135-37: Please clarify this sentence. 

Response: Thanks, we have modified as “This could be due to ensiling time that can provide high temperature and acidic conditions, thus detoxifying the HCN in FTMR [25].” 

Lines152-53: Please clarify this part of the sentence ‘……. was highest in the number of fibrolytic bacteria and anaerobic rumen fungi’ 

Response: We have modified as “McSweeney and Denman [29] reported sulfate-supplemented animals had significantly higher (29.6%) DM intake compared with the unsupplemented group, which might have influenced by increasing in the number of rumen microbes.” 

Line 156: It should be ‘fermentation in rumen’ 

Response: We have revised as suggestion. Please see in manuscript Line 156 

Line 182: Please expand ‘NH3-N’ 

Response: It was first described in section of Introduction. Please see in manuscript Line 48 

Lines 188-89: Please rewrite this sentence again. 

Response: We have modified as “Inclusion of a high level of urea with the supplement of an extremely fermentable carbohydrate source has improved dairy cows’ milk production. Please see in manuscript Line 188-189 

Lines 197-99: Please reword ‘The result was supported by the Uwituze et al. [36] research: Sheep fed a cassava-based was diet exhibited a blood thiocyanate concentration correlated with the level of sulfur’ as ‘This finding was supported by the findings of Uwituze et al. [36] in sheep that were fed a cassava-based diet and exhibited a blood thiocyanate concentration directly proportional to the level of sulfur supplementation’ 

Response: We have revised as suggestion. Please see in manuscript Line 197-199 

Line 246: Please clarify and rewrite this ‘Greater degradability of the result..’ 

Response: We have modified as “Greater degradability might be due to the better activity of microbial stimulate efficiently hydrolyzed to NH3 by urea.” Please see in manuscript Line 247 

Line 250: I think ‘different’ will be a better word than ‘Dissimilatory 

Response: We have revised as suggestion. Please see in manuscript Line 251 

Line 267: I think it should ‘Improvement in milk yield was not significant’ instead of ‘Milk yield was not significant’ 

Response: We have revised as suggestion. Please see in manuscript Line 268 

Lines 270-73: I am unable to comprehend these lines of the discussion on milk yield. Please clarify and rewrite ‘This yield was in the optimal range of tropical zone, which is typically associated with animal feeds and arrangement, animal genetics, and environment. This may also indicate the lack of significance compared with feed consumed among the animals; hence, no notable difference emerged for milk yield between the treatments. 

Response: Thanks for your comment. Now, we have removed these sentences and added some explanation accordingly to comment made by the Reviewer 1. Please see in manuscript Line 271 

Line 304: Please reword ‘Somatic cell count was decreased is possibly..’ as ‘The decrease in somatic cell count was  possibly…’ 

Response: We have revised as suggestion. Please see in manuscript Line 303 

Line 307: Please reword ‘to limit the scope somatic cell count’ as ‘to limit the increase in somatic cell count’ 

Response: We have revised as suggestion. Please see in manuscript Line 306 

Lines 313-15: Please clarify and rewrite this sentence again ‘ It prevents total flora and salmonella growth [51] and response to Streptococcus agalactiae subclinical mastitis, Staphylococcus  aureus and Escherichia coli which is the major bacteria of mastitis’ 

Response: In order to make it clearly more, now we have removed these sentence from manuscript.  

Thank you! 

Reviewer 2 Report

Reviewer comments for manuscript ID vetsci-855751 entitled ‘Fresh Cassava Root in Fermented Total Mixed Ration containing Sulfur and Urea Levels could Improve Milk Quality of Tropical Lactating Cows’

General comments

The authors have touched a growing field of animal nutrition research using locally available materials and their fortification with organic/inorganic compounds to improve animal productivity and performance. This research should be able to provide alternatives to traditional dairy feeding through cost cutting in production. I congratulate the authors for this work. I have some comments on the manuscript that I would like the authors to ponder and do the needful before recommendation for publication.

Introduction: Please elaborate on the gaps in literature you have identified and aim to fill through this research.

Results and discussion: This is overall a well written portion especially the discussion. However care must be taken at some points that I have specifically pointed out to properly clarify the points of discussion.

Conclusions and recommendations: Conclusions are too short and do not elaborate about the limitations of this research and future directions of the research. Polioencephalaomalacia due to sulphur supplementation as a passing reference in the end might put doubts in the applicability of this research. This point must be carefully dealt and elaborated. I found no recommendations.

Specific comments

Line 48: I could not understand this line. Please clarify ‘Non-protein nitrogen is used to analyze microbes in rumen because sulfur became limited. ‘

Line50-52: Please reframe ‘Sulfur and nitrogen are combined to improve the rumen microbes to synthesize amino acids, and digestion of cellulose could be attributable to activity of fiber fermentation [7]’ as ‘Sulfur and nitrogen are combined to improve the synthesize of amino acids, and digestion of cellulose by rumen microbes through higher activity of fiber fermentation [7]’

Line 54: Please clarify this sentence ‘…. which has efficient antimicrobial pathogens’

Lines 56-57: Please reframe ‘These have catalyzed the oxidation of thiocyanate by hydrogen peroxide with the synthesis of the antibacterial hypothiocyanite and other substrates’ as ‘These catalyze the oxidation of thiocyanate by hydrogen peroxide through the synthesis of the antibacterial hypothiocyanite and other substrates’

Line 59: Please replace ‘synthetic’ with ‘synthesized’

Line 62: Please reframe the sentence ‘It was hypothesized that fresh cassava root with sulfur and urea addition for ruminant diets and can also improve animal performance and feed efficiency’ as ‘It was hypothesized that fresh cassava root with the addition of sulfur and urea in ruminant diets improves animal performance and feed efficiency’

Line 68: Please replace ‘allowed’ with ‘approved’

Lines77-79: These sentences are contradictory ‘All dairy cows were fed in individual pens on an ad libitum basis. Animals were allowed free access to feed in the morning (07:00) and afternoon (16:00), while water and mineral block were provided by free choice’ Please clarify.

Lines 80-81: Please reword ‘….samples, rumen fluid, feces, and blood’ as ‘…… samples of rumen fluid, feces, and blood’

Lines 82-86: Please reframe ‘The FTMR was made according to Supapong et al. [12]. In short, the FTMR was made by packing concentrate mixed with rice straw chopped to a length of 3-5 cm by machine and adding water to achieve a moisture content of 55% in processing. FTMR was packed in plastic bags to ferment under anaerobic conditions and stored indoors (25-32 °C) for 7 days before use’ as ‘FTMR was made by packing concentrate mixed with rice straw chopped to a length of 3-5 cm by machine and adding water to achieve a moisture content of 55% in processing, as per the process carried out by Supapong et al. [12]. FTMR was then packed in plastic bags to ferment under anaerobic conditions and stored indoors (25-32 °C) for 7 days before use

Line 97: Please add concentration after ‘allantoin and creatinine’ to read as ‘….. allantoin and creatinine concentrations

Lines 110-112: Please rewrite the sentence again as ’50 ml of ruminal fluid sample was collected by using vacuum pump linked to a stomach tube at hour 0 and 4 h post feeding’

Line115: Please rewrite ‘The ruminal fluid 45 ml was added...’ as ‘45ml of ruminal fluid was added….’

Lines 126-27: Abbreviations used for the first time must be written in expanded form.

Lines135-37: Please clarify this sentence.

Lines152-53: Please clarify this part of the sentence ‘……. was highest in the number of fibrolytic bacteria and anaerobic rumen fungi’

Line 156: It should be ‘fermentation in rumen’

Line 182: Please expand ‘NH3-N’

Lines 188-89: Please rewrite this sentence again.

Lines 197-99: Please reword ‘The result was supported by the Uwituze et al. [36] research: Sheep fed a cassava-based was diet exhibited a blood thiocyanate concentration correlated with the level of sulfur’ as ‘This finding was supported by the findings of Uwituze et al. [36] in sheep that were fed a cassava-based diet and exhibited a blood thiocyanate concentration directly proportional to the level of sulfur supplementation’

Line 246: Please clarify and rewrite this ‘Greater degradability of the result..’

Line 250: I think ‘different’ will be a better word than ‘Dissimilatory’

Line 267: I think it should ‘Improvement in milk yield was not significant’ instead of ‘Milk yield was not significant’

Lines 270-73: I am unable to comprehend these lines of the discussion on milk yield. Please clarify and rewrite ‘This yield was in the optimal range of tropical zone, which is typically associated with animal feeds and arrangement, animal genetics, and environment. This may also indicate the lack of significance compared with feed consumed among the animals; hence, no notable difference emerged for milk yield between the treatments.

Line 304: Please reword ‘Somatic cell count was decreased is possibly..’ as ‘The decrease in somatic cell count was  possibly…’

Line 307: Please reword ‘to limit the scope somatic cell count’ as ‘to limit the increase in somatic cell count’

Lines 313-15: Please clarify and rewrite this sentence again ‘ It prevents total flora and salmonella growth [51] and response to Streptococcus agalactiae subclinical mastitis, Staphylococcus  aureus and Escherichia coli which is the major bacteria of mastitis’

Round 2

Reviewer 1 Report

Dear Authors,

I appreciate the clarifications made to the manuscript and especially in regards to my animal welfare concerns. Just as a final clarification on that topic, were these cows housed individually but in view of other cows? You mentioned a freestall environment, but does this mean multiple adjacent pens of cows in the same room so they had visual or auditory cues from other cows? Or simply a freestall with a rubber mat on it but no other cows around? If you would be able to clarify this it would be greatly appreciated.

If all cows were housed in the same room, this would significantly lessen the stress burden on the animals. The provided references I looked through however,  did not provide more clarity regarding the housing conditions.

I understand that nutritional studies involve far fewer animals compared to other animal based studies and I appreciate the clarified rationale, additional explanation of the facility environment and examples given of previously accepted studies with the similar methodology.

However, I would still prefer to have more of the animal and facility details in the manuscript, as provided in the answer to me as a reviewer, to ensure ample foundation for replicability by others without having to reach out to the authors in person. I.e., mentioning of the training, rubber mats, season and temperature span, pen size etc. It could even be its own sub-header under material and methods. This is simply a recommendation from my side and not a requirement for publication, but as international researchers might be under different restrictions or guidelines for nutritional studies and animal housing, it could be helpful to provide. Please consider.

Finally, my main concern regarding the emphasis on the derived results based on few animals has now been reworded more cautiously.

Line by line comments:

Ln 39-67. I see no direct additions to the introduction per reviewer 2’s comment on gaps in literature. Consider adding a sentence or two on what additional knowledge that can be gained by this particular study.

Ln 78-79:  I still think that this sentence needs some clarification. Please correct me if I understood this incorrectly.

If feed was left out but replenished later in the day after refusals were collected and weighed, consider revising to: “All cows had ad libitum access to FTMR that was provided fresh twice daily (07:00 and 16:00), allowing for 10% refusals. Refusals were collected and weighed each morning and afternoon before fresh feed was provided.”

Also, here and elsewhere, avoid starting a sentence with an abbreviation. (E.g. line 83, 126, 127,184,189, 194 and possibly elsewhere).

Ln 302-310. The answers provided to me as a reviewer makes perfect sense. However, none of these controlled for factors are mentioned here or in the material and methods section of the manuscript. Thus, adding this information previously would make the omission of this discussion in this section more appropriate. Otherwise it could come across as overlooked when in fact they were controlled for at the research center.

Reviewer 2 Report

General comments

I appreciate the authors for undertaking a thorough revision of the manuscript to improve its quality. I have few minor comments/corrections to suggest. I recommended the publication of the manuscript.

Specific comments

Title: I suggest a modification in the title as ‘Effect of sulfur and urea fortification of Fresh Cassava Root in Fermented Total Mixed Ration on the Improvement Milk Quality of Tropical Lactating Cows’

Line 49: Replace ‘microorganism’ with ‘microorganisms’

Line 51: Replace ‘synthesize’ with ‘synthesis’
